# Polyphasic Characterization of *Geotalea uranireducens* NIT-SL11 Newly Isolated from a Complex of Sewage Sludge and Microbially Reduced Graphene Oxide

**DOI:** 10.3390/microorganisms11020349

**Published:** 2023-01-30

**Authors:** Li Xie, Naoko Yoshida, Lingyu Meng

**Affiliations:** Department of Civil Engineering, Nagoya Institute of Technology (Nitech), Nagoya 466-8555, Japan

**Keywords:** *Geotalea*, electrogenic bacteria, complete genome, graphene oxide

## Abstract

Graphene oxide (GO), a chemically oxidized sheet of graphite, has been used as a conductive carbon carrier of microbes to boost various bioelectrochemical reactions. However, the types of microbes that can reduce GO have rarely been investigated. In this study, a strain of GO-reducing bacteria, named NIT-SL11, which was obtained from a hydrogel of microbially reduced GO and anaerobic sludge that converts sewage to electricity, was phylogenically identified as a novel strain of *Geotalea uraniireducens*. Considering the current lack of information on the electrogenic ability of the bacterium and its physicochemical and chemotaxonomic characteristics, the polyphasic characterization of the *Geotalea uraniireducens* strain NIT-SL11 was performed. NIT-SL11 utilized various organic acids, such as lactate, benzoate, and formate, as electron donors and exhibited respiration using GO, electrodes, fumarate, and malate. The strain contained C16:1ω7c and C16:0 as the major fatty acids and MK-8 and 9 as the major respiratory quinones. The complete genome of NIT-SL11 was 4.7 Mbp in size with a G+C content of 60.9%, and it encoded 80 putative c-type cytochromes and 23 type IV pili-related proteins. The possible extracellular electron transfer (EET) pathways of the strain were the porin–cytochrome (Pcc) EET pathway and type IV pili-based pathway.

## 1. Introduction

Bioelectrochemical systems, including microbial fuel cells (MFCs) and microbial electrolysis cells (MECs), and their applications in wastewater treatment have received much attention [1]. In MFC wastewater treatment, microorganisms degrade organic matter and transfer electrons to the anode; the electrons recovered in the anode are later utilized to reduce oxygen at the cathode. Thus, the anode is a critical factor that affects the energy recovery in bioelectrochemical systems, especially in the case of wastewater with a low chemical oxygen demand [2,3]. Preferably, the anode exhibits chemical stability, good affinity for microbes, and a large surface area [4,5,6]. Thus, various 3D anodes have been applied to facilitate current recovery [7,8,9,10]. 

The use of graphene oxide (GO), the oxidized form of graphene, has facilitated electricity production in several MFCs via microbial the reduction of non-conductive GO to conductive reduced GO (rGO) [11]. GO promotes the growth of selective exoelectrogens [12,13] by serving as the extracellular electron acceptor and self-aggregating into a 3D-conductive hydrogel that embeds exoelectrogens and results in significantly stabler energy production than is the case with graphite felt (GF) [14] and electrochemically oxidized GF [15]. Accordingly, hydrogels have been applied in the recovery of electricity from wastewater [14,16,17], including a relatively large-scale (100 L) swine wastewater treatment system utilizing MFCs [18]. To date, various microorganisms have been reported to reduce GO, including the following genera: *Shewanella* [11], *Escherichia* [18], *Citrifermentans* [12,19], and *Desulfuromonas* [13,20], as well as taxa found in natural microcosms [21]. However, exoelectrogens in the rGO complex converting sewage to electricity have yet to be isolated and characterized. In this study, an electrogenic bacterium, designated NIT-SL11, was successfully isolated from the rGO complex that converts sewage to electricity and is described in detail. NIT-SL11 is capable of growing under anaerobic conditions by oxidizing acetate coupled with Fe(III) reduction and is phylogenetically identified as a strain of *Geotalea uraniireducens*. The potential extracellular electron-transferring (EET) pathway is also proposed based on the genome sequence.

## 2. Materials and Methods

### 2.1. Isolation and Growth Conditions of NIT-SL11

Strain NIT-SL11 was isolated from an rGO complex anode utilized to recover electricity from municipal sewage wastewater, as previously described [14]. Microbes in the rGO anode were cultivated on an anaerobic agar plate supplemented with 5.0 mM acetate, 5.0 mM disodium anthraquinone-2,6-disulfonate (AQDS-Na), and 10% sludge extract. The agar plate was composed of DHB-CO_3_AAQ medium, which is based on a DHB-CO_3_ medium [22] with the following modifications: replacement of GO with 5.0 mM AQDS and supplementation with 0.5% agarose. After 7–14 days of incubation of the plate at 28 °C, electrochemically active colonies were visible due to orange halos, which is the color of the reduced form of AQDS. The colony culture was then purified by repeated agar-shake cultivation using DHB-CO_3_-AFY medium, which is a modified DHB-CO_3_ medium supplemented with 5.0 mM acetate, 5.0 mM fumarate, 0.1% yeast extract, and 0.1% yeast extract. The colony was collected from the agar culture, purified by repeated agar-shaking cultivation, and then re-cultivated in liquid DHB-CO_3_-AF medium, that is, DHB-CO_3_ medium supplemented with 5.0 mM acetate and 5.0 mM fumarate. Lastly, based on the uniformity of the microscopic morphology and 16S rRNA gene sequences, one of these liquid cultures was selected and further purified by repeating the agar cultivation step. The purified culture was then phylogenetically identified based on sequencing of the 16SrRNA gene amplified from the cell lysate [23] and named strain NIT-SL11. Strain NIT-SL11 was routinely cultured in liquid DHB-CO_3_-AF. In total, 7–14 days of anaerobic cultivation at 28 °C were sufficient to achieve full growth of the bacterial colony. 

### 2.2. Morphological, Physiological, and Biochemical Analyses 

The morphology of strain NIT-SL11 was observed under a field-emission scanning electron microscope (JSM-7800F; JEOL Ltd., Tokyo, Japan) operating at 1.0 kV [19], and its spore-forming ability and Gram-stainability were checked using optical microscopy, as previously described [24]. The salinity, temperature, and pH tolerance of NIT-SL11 were evaluated by measuring the growth of cells in DHB-CO_3_-AF medium. Salinity tolerance was examined by supplementing with 0 to 8% (*w*/*v*) of NaCl. The pH of the bicarbonate-free medium was adjusted to the range of 5.2 to 8.6 by the addition of sodium bicarbonate in the medium and CO_2_ in the headspace for the pH tolerance tests. The cell growth at different temperatures was tested in the range between 4 °C and 10 to 40 °C, with approximately 5 °C intervals.

The ability of strain NIT-SL11 to utilize an e^−^ donor was determined by observing cell growth with the following substances in combination with reduction of 5 mM fumarate: 10 mM of formate, acetate, butyrate, lactate, pyruvate, succinate, propionate, malate, isobutyrate, caproate, benzoate, phenol, methanol, isopropanol, ethanol, butanol, glucose, fructose, and glycerol, and 0.5 g/L of peptone and yeast extract. Potential electron acceptors used by strain NIT-SL11 were assayed by observing cell growth in 10 mM each of fumarate, malate, sulfate, and thiosulfate; 5 mM of AQDS and nitrate; and 20 g/L of elemental sulfur with oxidation of 5 mM acetate. The production of electric current by the strain NIT-SL11 was evaluated via electrochemical cultivation using a graphite plate inoculated with NIT-SL11, as previously described [12].

### 2.3. Chemotaxonomic Analysis 

The isoprenoid quinones and cellular fatty acid composition were investigated by Techno Suruga Laboratory Co., Ltd. (Shizuoka, Japan), as described in a previous study [24]. Isoprenoid quinones were extracted as previously described by Tamaoka et al. [25]. The analysis of cellular fatty acids was carried out using cells grown in liquid DHB-CO_3_-AF medium at 28 °C for 14 days, and the fatty acid profile was obtained according to the protocol of the Sherlock Microbial Identification System v6.0 (MIS, Newark, DE, USA) by accessing the TSBA6 database.

### 2.4. Genetic Characterization

The genomic DNA of strain NIT-SL11 was extracted as previously described [26,27]. Sequencing was performed using a combination of Illumina and Nanopore sequencing reads. A total of 1.8 M reads (1.02 GMbp) of paired-end reads and 0.37 M reads (1.76 GMbp) of single reads were subjected to error removal using Short Read Manager and assembled using Unicycler (v0.4.7, https://github.com/rrwick/Unicycler/releases/tag/v0.4.7, accessed on 18 October 2021). Gap sequences were determined in silico using Geno Finisher (v7.0, in silico biology, inc. Yokohama, Japan) [28]. Gene prediction and annotation of the complete genome of strainNIT-SL11 were performed using DFAST (https://dfast.ddbj.nig.ac.jp/, accessed on 11 October 2021) [29]. Gene comparison of NIT-SL11 and other *Geobacter* species was based on bidirectional best hits at 40% identity and 80% query coverage by SEED Viewer v2.0 (https://rast.nmpdr.org/rast.cgi, accessed on 18 October 2021) [30] and BLAST (https://blast.ncbi.nlm.nih.gov/Blast.cgi, accessed on 26 October 2021) in NCBI database. The complete genome sequence of strain NIT-SL11 was deposited in DDBJ/GenBank under the BioProject number PRJDB15015 with accession number AP027151.

The full-length 16S rRNA gene sequence of strain NIT-SL11 was extracted from the complete genome data, and those for all publicly available *Geobacter* spp. were downloaded from the NCBI database. Phylogenetic analyses were performed using MEGA X (v8, Max Planck Institute of Biochemistry (MPIB), Planegg, Germany) (https://www.megasoftware.net/, accessed on 18 October 2022) based on the neighbor-joining method [31]. Statistical support for the branches of the phylogenetic trees was determined using bootstrap analysis based on 1000 re-samplings [32].

## 3. Results and Discussion

### 3.1. Isolation of NIT-SL11

The NIT-SL11 colonies were successfully cultivated following the method described above (Section 2.2). The cells of strain NIT-SL11 were found to be Gram-negative, non-spore-forming, rod-shaped, and approximately 0.4 µm in width and 1.4 µm in length (Figure 1A).

In the electrochemical cultivation, the colonies of strain NIT-SL11 grew and generated an orange-colored biofilm (Figure 1B) on a graphite electrode supplemented with acetate. Simultaneously, an electric current was rapidly generated and reached its maximum, 0.55 mA/cm^2^, on day 8 (Figure 1C). The current decreased over time but increased immediately after the addition of acetate to the medium. These results demonstrate that NIT-SL11 grew by coupling extracellular electron transfer with the electrode via acetate oxidation.

### 3.2. Phylogenetic Identification Based on 16S rRNA Sequencing

Strain NIT-SL11 was found to contain two 16S rRNA operons (rrn1 and 2), and the 16S rRNA gene-based phylogenetic tree revealed that the two rrns formed a cluster with *Geotalea uraniireducens* Rf4^T^ (Figure 2). The 16S rRNA gene sequence similarity was 98.12% to 98.17% with respect to the other members of the genus *Geotalea*. Based on the cut-off values of 98.2–99.0% [33], and the 98.65% [34] similarity within the single species, the strain NIT-SL11 can be identified as a novel strain of *G. uraniireducens*. The genus *Geotalea* was recently proposed by Lovely et al. [35] by dividing *Geobacter* into four genera: *Geobacter*, *Trichlorobacter*, *Citrifermentans*, and *Geotalea* [36]. The genus *Geotalea* includes three species: *G. uraniireducens* [37] and *G. daltonii* [38], which were isolated from subsurface sediments, and *G. toluenoxydans*, which was isolated from the oil of a former coal-gasification site [39]. The isolation of a novel strain of *G. uraniireducens* from sewage wastewater indicated the widespread distribution of *Geotalea* in various environments. *Geotalea* species are Gram-negative, non-spore-forming rods, which are obligately anaerobic and typically able to oxidize acetate via coupling with fumarate reduction, although the potential for e- donor and acceptor utilization has not been investigated in this genus (Table 1). To the best of our knowledge, the present study is the first to demonstrate electricity production by a pure strain of *Geotalea*.

### 3.3. Physiological and Biochemical Characterization

Strain NIT-SL11 was found to tolerate temperatures in the range of 20 to 35 °C (optimum at 25–30 °C), pH 5.6–8.3 (optimum at 6.0–6.7), and NaCl concentrations of 0 to 2% (with an optimum at 0–1%). As shown in Table 1, consistent with most *Geotalea* strains, NIT-SL11 utilized acetate, benzoate, formate, and pyruvate as e^−^ donors, with fumarate as the electron acceptor. Moreover, NIT-SL11 grew with the oxidization of lactate, peptone, pyruvate, and yeast extract, but not butanol, butyrate, caproate, ethanol, fructose, glucose, glycerol, isobutyrate, isopropanol, malate, methanol, phenol, propionate, and succinate. Strain NIT-SL11 was able to reduce fumarate and malate using acetate as an electron donor, whereas AQDS was not utilized as an electron acceptor. Additionally, NIT-SL11 was unable to reduce elemental sulfur, sulfate, and thiosulfate, similar to most *Geotalea* strains. None of the *Geotalea* strains were able to reduce nitrate levels. The cells of strain NIT-SL11 did not show any apparent movement on the slide, suggesting that strain NIT-SL11 was non-motile.

### 3.4. Chemotaxonomic Characterization

To the best of our knowledge, the major respiratory quinones and cellular fatty acids of *G. uranireducens* have not been reported. Therefore, such an analysis was performed in the present study using NIT-SL11. As shown in Table 2, the predominant quinones of strain NIT-SL11 were MK-8 (94%) and MK-9 (6.3%), and the major quinone was consistent with *G. toluenoxydans* TMJ1^T^. The predominant cellular fatty acids of strain NIT-SL11 were C_16:1_ω7*c* (39%) and C_16:0_ (32%), similar to those of *G. daltonii* FRC-32^T^ and *G. uraniireducens* Rf4^T^, but not *G. toluenoxydans* TMJ1^T^, for which the main cellular fatty acids are C_16 : 0_ and C_18 : 0_.

### 3.5. General Genomic Features

An overview of the genome of strain NIT-SL11 and a comparison with other strains of the genus *Geotalea* are shown in Table 2. The genome size of strain NIT-SL11 was estimated to be 4.19 Mb, which approximates that of *G. toluenoxydans* TMJ1^T^ and *G. daltonii* FRC-32^T^, whereas *G. uraniireducens* Rf4^T^ showed a relatively larger genome size of approximately 5 Mb. The genome of NIT-SL11 encodes 3860 CDSs, 56 tRNAs, and 6 rRNAs. The G+C content of strain NIT-SL11 was calculated as 63.1%, which is relatively higher than that of other *Geotalea* strains (Table 2).

A genome map is shown in Figure 3. The KEGG category revealed that the highest number of genes were related to the metabolism of amino acids and their derivatives (200 genes), followed by protein metabolism (141 genes) and carbohydrate metabolism (135 genes). Regarding energy conversion, NIT-SL11 was found to have CDSs, which metabolize organic acids, such as formate, lactate, and pyruvate. The strain also showed a gene set for the complete TCA cycle, including CDSs, which metabolize organic acid intermediates (fumarate and malate) of the TCA cycle. NIT-SL11 also contains a full set of genes associated with glycolysis; however, the bacteria cannot utilize glucose because of the lack of glucose transporters. For nitrogen metabolism, NIT-SL11 only had CDSs for nitrogen fixation but not for the assimilation and dissimilation of nitrate reduction and denitrification. The sulfur metabolism pathway of NIT-SL11 was incomplete due to the absence of reductase-related genes for sulfur and thiosulfate. NIT-SL11 contained a full set of genes for assimilatory sulfate reduction; however, it lacked an extracellular sulfate transport system substrate-binding protein. These results are consistent with the inability of NIT-SL11 to use sulfur, sulfate, and thiosulfate as electron acceptors (Table 1).

### 3.6. Putative c-Type Cytochromes 

NIT-SL11 was found to possess 80 putative *c*-type cytochromes (Table 3) via sequence screening, as previously described [20]. Based on PROSITE prediction [40], most c-type cytochromes were present in the periplasmic (19), extracellular (14), outer (9), and cytoplasmic membranes (5), whereas few cytochromes were present in the cytoplasm (3) and inner membrane (1). The number of heme-binding motifs varied from one to twenty-seven.

Seventy-three c-type cytochromes are homologs of those functionally identified in *Geobacter sulfurreducens* PCA, a well-characterized model strain of *Geobacter* genus. GURASL_06470 and 18290 are homologous to CbcA; GURASL_02110, 32780, 32770, 32730, and 22490 are homologous to CbcL, CbcM, CbcN, CbcR, and CbcX, respectively; and GURASL_ 03150 is homologous to ImcH. GURASL_06660 and 03230 are homologous to PpcA, and GURASL_03240 and 21440 are homologous to PpcC and PpcE, respectively. GURASL_08460 and 28660 are homologous to OmaB, and GURASL_28710, 28700, and 28650 are homologous to OmcB. These cytochromes are involved in the porin–cytochrome (Pcc) EET pathways that transfer electron across the cell envelope [41], indicating that the Pcc pathway may be one of the major EET pathways in strain NIT-SL11 (Figure 4). The Cbc complexes and ImcH are inner-membrane cytochromes that oxidize quinol in the cytoplasmic membrane and transfer the released electrons to the periplasmic PpcA homologs [42,43], which transfer the electrons acquired from the cytoplasm to the OmcB-based outer-membrane complex (ombB-omaB-omcB). The OmcB-based conduit transfers electrons through the lipid bilayer of the proteoliposomes [44,45]. It has been reported that, in the case of *Geobacter sulfurreducens*, CbcL is required for the reduction of electron acceptors with reduction potentials at or below −100 mV, and ImcH is necessary for the reduction of electron acceptors with reduction potentials above −100 mV [20]. Thus, there may also be two different pathways in NIT-SL11 grown on electrodes poised at different oxidizing potentials—the CbcL-dependent and ImcH-dependent pathways (Figure 4)—by which electrons are transported out of the inner-membrane quinone pool.

Strain NIT-SL11 also contains homologs of several other outer-membrane cytochromes in *Geobacter sulfurreducens*: GURASL_06720 and 17250 are homologous to OmcE, whereas GURASL_17360 is homologous to OmcS, which is suggested to transfer electrons to the T4P apparatus [46]; GURASL_32490, 32200, and 32150 are homologous to OmcG; GURASL_32160, 32170, 18110, and 32550 are homologous to OmcH; GURASL_26970 and 32480 are homologous to OmcM; and GURASL_27550, 17570, 15180, 32470, 32560, 06450, 19400, 07200, and 28530 are homologous to OmcF, I, K, N, P, Q, V, X, and Y, respectively. Most of these Omc-proteins have been reported to be involved in Fe(III) reduction of *Geobacter* species [47,48] and are upregulated during the growth of *G. uraniireducens* on Fe(III) oxides and/or Mn(IV) oxides [47]. GURASL_30840, 30810, 09840, 32800, 17430, and 17440 are homologs of ExtA, D, G, K, Q, and R, respectively. These genes belong to the outer-membrane conduit *ext* cluster, which aid electron transfer across the outer membrane to the bacterial surface [49]. Additionally, GURASL_22590 is homologous to the periplasmic electron transfer protein PgcA, which facilitates respiration to Fe(III) oxides but not to electrodes [50]. GURASL_ 35130 is a homolog of NrfA, which catalyzes the reduction of nitrite to ammonium via the dissimilatory nitrate reduction to ammonium (DNRA) pathway [51].

### 3.7. Type IV Pilus (T4P)-Related Genes

Strain NIT-T3 contains 23 CDSs encoding T4P (Table 4). T4P are filamentous polymers of pilin monomers that undergo dynamic rapid polymerization and depolymerization from a pool of pilins [52]. The NIT-SL11 genome contained a full set of genes encoding T4P: one major pilin PilA, one minor pilin PilE, and other essential proteins for secretin (PilQ), alignment (PilM, O, and P), platform (PilC), retraction ATPases (PilT), and assembly ATPases (PilB). All CDSs were found to be well conserved in the *Geobacter* strains. The pilus polymer pilin protein PilA of *G. sulfurreducens* is an electrically conductive pilus that is known to be involved in the EET of solid electron acceptors [53]. Aromatic acids are key factors associated with conductivity. The aromatic acid content in the PilA homolog in NIT-SL11 was estimated to account for 8.45%, which is consistent with the range of PilA aromatic acid content in phylogenetically diverse bacteria (5.5–25.25%) [54]. This suggests that the polymer was conductive. Thus, T4P transfers electrons directly to oxides distant from cell surfaces, which may be the second EET pathway of strain NIT-SL11 (Figure 4).

### 3.8. Exoelectrogens That form the rGO Complex

The type of electrode material is a crucial determinant of the formation of biofilms and the performance of electron transfer at the cell–electrode interface, which affects electricity production in bioelectrochemical systems. Many attempts have been made to propose novel electrode materials (e.g., carbon brush, carbon fabric, and GO) for use in such systems, and improved performance has been confirmed [55]. As a promising electrode material, GO electrodes have shown stabler energy production than conventional electrode materials (GF), mainly due to the formation of a self-aggregated conductive hydrogel (rGO complex) by the interaction between exoelectrogens and GO [18]. However, not all exoelectrogens can interact with GO to form a hydrogel [12], and the mechanism of hydrogel formation is poorly understood. To date, it has been shown that two exopolysaccharide (EPS) components, alpha-polysaccharides and bGlcNAc polysaccharides, are key players in the hydrogel formation of *Shewanella* BC01 and CN32 [56], which can be generated during bacterial growth, are readily adsorbed by rGO, and are helpful in terms of the in situ gelling of the bacteria/GO complex [57,58]. However, the most-studied strain, *S. oneidensis* MR-1, cannot form a hydrogel due to the lack of relevant genes [58]. On the other hand, although constituting another representative exoelectrogen, studies regarding hydrogel formation by *Geobacter* are limited. These genes are not coded in only two of the known hydrogel-forming *Geobacter* species, *C. bremensis* R4 [19] and G. NIT-SL11, which indicates that *Geobacter* and *Shewanella* differ in terms of the mechanism of hydrogel formation. Given that *Geobacter* is more predominant in MFC for wastewater treatment, the isolation of strain NIT-SL11 expands our understanding of the hydrogel formation mechanism of *Geobacter*. Further studies should focus on clarifying the functional genes and interaction between *Geobacter* and GO, which could form the basis for developing novel and effective electrode materials and would further advance the applications of MFC.

## 4. Conclusions

In this study, *Geotalea uraniireducens* NIT-SL11 was obtained from a hydrogel of microbially reduced GO and anaerobic sludge. The isolated strain was found to utilize various organic acids as electron donors and respired with GO, electrodes, fumarate, and malate. The analysis of the genome of NIT-SL11 suggests two possible extracellular electron transfer pathways. In the first possible pathway, porin–cytochrome (Pcc) EET pathways that start with the transfer of electrons from the inner membrane quinone pool to the membrane-associated cytochromes Cbcl or ImcH, which transfer these electrons further to periplasmic electron carrier PpcA homologs, and the PpcA homologs transfer the electrons from the cytoplasm to OmcB-based or *Ext*-cluster outer-membrane conduits, which, finally, transfer electrons through the outer membrane. In the second possible pathway, the aromatic amino acid-rich conductive pili transfer electrons directly to oxides distant from the cell surfaces.

## Figures and Tables

**Figure 1 microorganisms-11-00349-f001:**
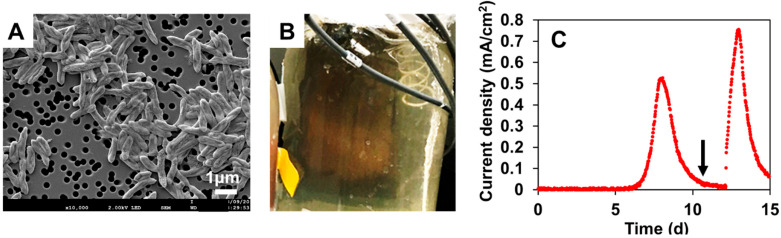
Morphology and electrogenic properties of strain NIT-SL11. (**A**) Scanning electron microscopic image of strain NIT-SL11. (**B**) Biofilm generated on an electrode in an electrochemical culture. (**C**) Electric current production by strain NIT-SL11. The arrow in panel C indicates a spike in acetate addition.

**Figure 2 microorganisms-11-00349-f002:**
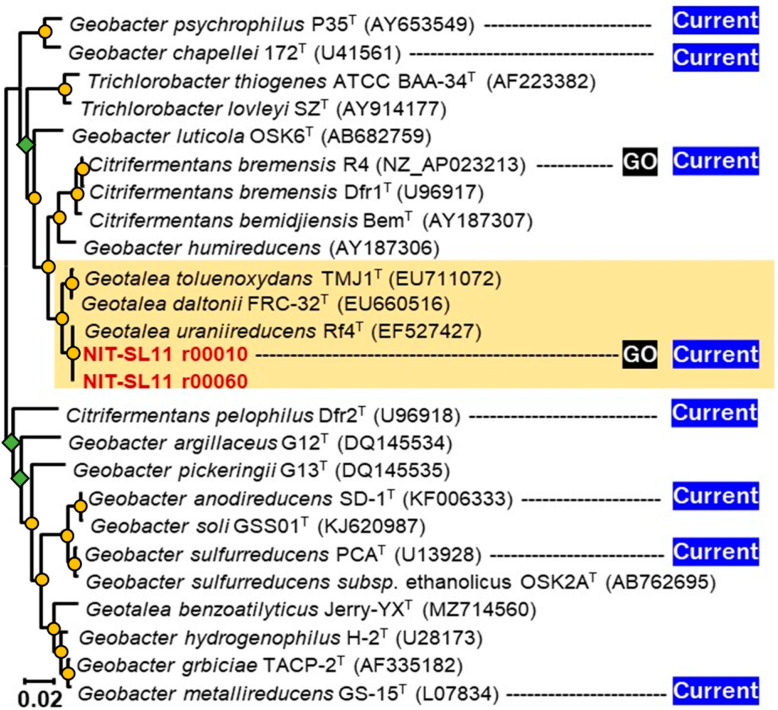
Phylogenetic tree generated using 16S rRNA gene sequences of members of Geobacteraceae family. Yellow circles and green diamonds indicate bootstraps >80% and <60%, respectively. GenBank accession numbers are stated in parentheses; GO: the ability to respire graphene oxide; Current: ability to generate current.

**Figure 3 microorganisms-11-00349-f003:**
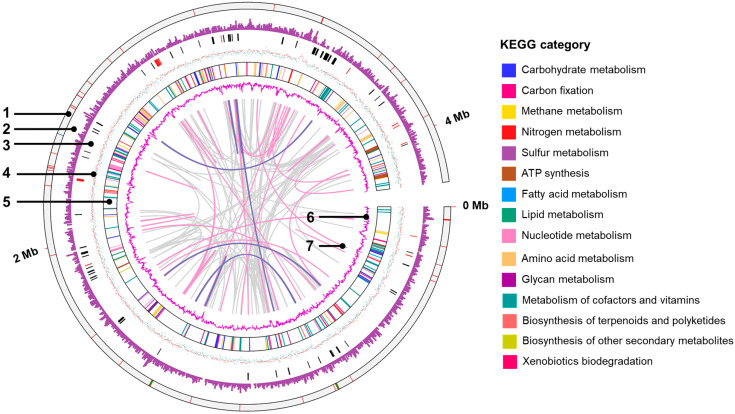
Features of the complete genome of strain NIT-SL11. Circular representation of the genome was generated using Tbtools-Ⅱ (v1.106, https://github.com/CJ-Chen/TBtools-Manual). Rings numbered from the outside to inside are: 1, location of tRNA (red), Transfer messenger RNA (blue), and rRNA (green); 2, gene density; 3, c-type cytochromes (black) and type IV pili (red); 4, G+C skew (red, positive; green, negative); 5, protein-coding sequences colored based on KEGG category; 6, G+C content; and 7, links showing repetitive sequence ≥95% identity (pink, >500 bp; purple, >2 kbp). KEGG, Kyoto Encyclopedia of Genes and Genomes.

**Figure 4 microorganisms-11-00349-f004:**
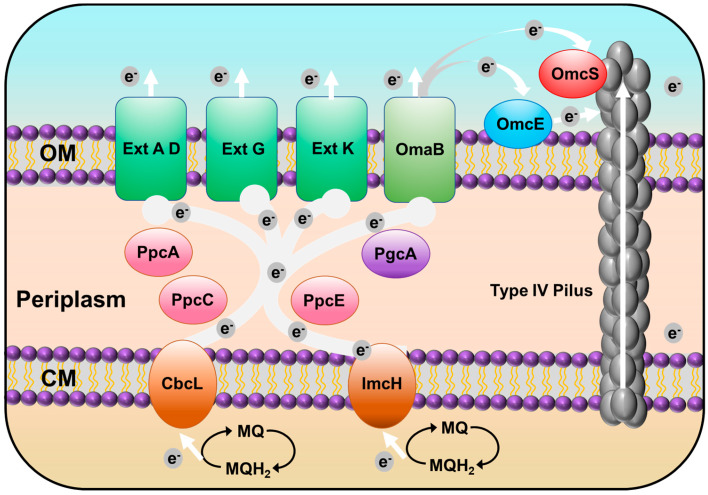
Proposed EET pathway of strain NIT-SL11. OM: outer membrane; CM: cytoplasmic membrane.

**Table 1 microorganisms-11-00349-t001:** Comparison of morphological and physiological properties of strain NIT-SL11 and recognized species of the genus *Geotalea*.

	Strain NIT-SL11	*G. toluenoxydans* TMJ1^T^	*G. daltonii* FRC-32^T^	*G. uraniireducens* Rf4^T^
Size (µm)	0.4 × 1.4	0.4 × 2.1–3.8	0.3–0.5 × 1.0–1.5	0.5–0.6 × 1.2–2.0
Motility	−	−	ND	+
Optimum Temp. (°C)	25–30	25–32	30	32
Optimum pH	6.0–.7	6.6–7.0	6.7–7.3	6.5–7.0
NaCl (%)	0–2	ND	0–0.7	ND
e^−^ donor				
Butanol	−	ND	+	−
Butyrate	−	+	+	−
Caproate	−	ND	ND	ND
Ethanol	−	ND	ND	+
Fructose	−	ND	ND	ND
Glucose	−	ND	ND	ND
Glycerol	−	ND	ND	ND
Isobutyrate	−	ND	ND	ND
Isopropanol	−	ND	ND	ND
Malate	−	−	ND	ND
Methanol	−	ND	ND	−
Phenol	−	+	ND	ND
Propionate	−	+	−	−
Succinate	−	−	−	−
Acetate	+	+	ND	+
Benzoate	+	+	+	ND
Formate	+	+	+	−
Lactate	+	−	−	+
Peptone	+	ND	ND	ND
Pyruvate	+	+	ND	+
YE	+	ND	ND	ND
H_2_	ND	−	−	−
e− acceptors				
Nitrate	−	−	−	−
Sulfate	−	−	ND	−
Thiosulfate	−	−	ND	−
AQDS	−	ND	ND	+
S^0^	−	−	+	−
Fumarate	+	+	+	+
Malate	+	ND	+	+
GO	+	ND	ND	ND
Electrode	+	ND	ND	ND
Fe(III)	ND	+	+	−
Mn(Ⅳ)	ND	−	ND	ND

ND: data not obtained. T: typical strain.

**Table 2 microorganisms-11-00349-t002:** Comparison of chemotaxonomic and genomic properties of strain NIT-SL11 and recognized species of the genus *Geotalea*.

	Strain NIT-SL11	*G. toluenoxydans* TMJ1^T^	*G. daltonii* FRC-32^T^	*G. uraniireducens* Rf4^T^
Size (Mb)	4.2	4.2	4.3	5.1
GC (%)	60	54	53	54
Total predicted genes	3923	−	3852	4591
CDSs	3860	−	3745	4457
rRNA	6	−	6	6
tRNA	56	−	49	49
CytC	80	−	73	69
Menaquinone	8, 9	8	−	−
Fatty acids >40%	−	C_16:0_	−	−
30–40%	C_16:1_ω7*c* , C_16:0_	−	C_16:1_ω7*c* , C_16:0_	C_16:1_ω7*c*
20–30%	−	C_18:0_	−	C_16:0_
10–20%	−	−	ios-C_15:0_	ios-C_15:0 ,_C_14:0_

T: typical strain.

**Table 3 microorganisms-11-00349-t003:** List of putative c-type cytochrome proteins present in the NIT-SL11 genome.

Tag	Local	CXnCH	1	2	3
n = 2	n = 3	n = 4
01590	−	3	0	0	0105	−	4255
02110	CM	8	0	0	cbcL,0274	3686	0125
03150	CM	6	1	0	imcH,3259	1014	0861
03230	−	3	0	0	ppcA,0612	1426	4121
03240	PP	3	0	0	ppcC,0365	1426	4121
06040	−	3	0	0	0533	1401	3909
06440	CP	2	0	1	0591	1590	3656
06450	OM	12	0	0	omcQ,0592	2170	3655
06470	PP	7	0	0	cbcA,0594	3403	2712
06660	PP	3	0	0	ppcA,0612	1426	4121
06690	PP	9	2	0	0615	1429	3840
06700	OM	8	0	0	ctcB,0616	1430	3839
06720	OM	4	0	0	omcE,0618	1685	3837
07200	PP	12	0	0	omcX,0670	0830	0641
07560	−	5	0	0	extCF,2725	1685	1838
08460	−	8	0	0	omaB,2738	1681	0988
08470	−	9	0	0	omcB,2737	1682	0989
08960	PP	2	0	0	imcG,1538	1685	1316
09830	−	5	0	0	extCF,2725	1685	1838
09840	EX	12	1	1	extG,2724	−	1837
13150	−	1	0	0	1284	2256	2997
15110	−	27	0	0	2210	2170	3135
15170	OM	1	0	0	2204	−	−
15180	PP	10	0	0	omcK,2203	1685	2291
15200	−	8	0	0	cytT,2299	2175	0672
17180	CM	1	0	0	coxB,0222	−	0414
17250	EX	4	0	0	omcE,0618	1685	3837
17260	EX	8	0	0	2076	−	−
17360	−	6	0	0	omcS,2503	3166	1989
17430	PP	26	0	0	extQ,2495	3160	1995
17440	PP	16	0	0	extR,2494	3159	1996
17570	PP	9	1	0	omcI,1228	2912	2291
17990	PP	2	0	0	macA,0466	2579	1316
18080	CM	8	0	0	2076	−	−
18110	−	13	0	1	omcH,2884	0831	1834
18290	CM	7	0	0	cbcA,0594	3403	2712
19400	PP	12	0	0	omcV,1996	2379	2822
21440	PP	3	0	0	ppcE,1760	1426	3843
21680	PP	1	0	0	cycC,1740	−	2136
22490	−	5	0	0	cbcX,1648	−	2381
22590	EX	2	0	0	pgcA,1761	3176	2022
22840	−	3	1	0	ctcD,1785	−	−
22850	CP	5	1	0	1786	2641	1922
22860	−	4	0	0	1787	2640	1921
25430	IM	1	0	0	ccoP,2513	−	1913
25830	−	3	0	0	3214	1685	3748
26520	EX	8	0	0	2076	−	−
26970	EX	6	0	0	omcM,2294	−	−
27550	OM	1	0	0	omcF,2432	−	0331
28090	EX	14	0	3	omcN,2898	3132	2035
28530	OM	8	0	0	omcY,2201	2175	3130
28650	EX	10	0	0	omcB,2737	1861	0989
28660	−	8	0	0	omaB,2738	1681	0988
28700	EX	12	0	0	omcB,2737	1682	0994
28710	−	8	0	0	omaB,2738	1681	0988
28780	PP	1	0	0	petJ,2743	−	0331
29020	OM	14	0	3	omcN,2898	3132	2035
30810	−	5	0	0	extD,2642	−	−
30820	−	4	0	1	extCF,2643	1681	1838
30840	PP	11	1	0	extA,2645	1685	0641
31410	OM	5	0	0	ctcC,2801	0309	−
32150	−	20	0	4	omcG,2882	3132	3427
32160	−	19	0	4	omcH,2883	3132	3427
32170	−	24	0	3	omcH,2884	1685	3428
32200	−	21	0	5	omcG,0702	1685	2035
32390	OM	23	0	3	omcN,2898	1513	3428
32470	EX	20	0	3	omcN,2898	1513	2035
32480	EX	15	0	8	omcM,2899	1685	0076
32490	−	20	0	7	omcG,0702	1513	3427
32550	EX	23	1	3	omcH,2912	1685	3428
32560	EX	4	0	0	omcP,2913	1432	3837
32700	−	2	0	0	2927	1359	4833
32730	−	4	0	0	cbcR,2930	1348	0460
32770	CP	10	0	0	cbcN,2934	1344	0456
32780	EX	12	0	0	cbcM,2935	1343	−
32800	PP	5	0	0	extK,2937	1340	0447
34580	PP	9	0	0	3137	1724	−
34890	−	2	0	0	2767	1685	4149
35130	PP	4	0	0	nrfA,3154	3111	0665
37130	−	2	0	0	3332	3818	4408
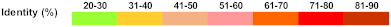

1: *Geobacter. sulfurreducens* PCA1; 2: *G. daltonii* FRC-32; 3: *G. uraniireducens* Rf4. Tag indicates the locus tag of the coding sequence encoding c-type cytochromes in the genome. CM, cytoplasmic membrane; PP, periplasm; EX, extracellular; (−), unknown. The number and color scale for the closest protein in the strain indicate the locus tag and amino acid identity (%), respectively.

**Table 4 microorganisms-11-00349-t004:** List of T4P assembly-related genes in the NIT-SL11 genome.

Tag	Annotation				Aromatic Acid Mole %	1	2	3
20580	geopilin domain 1 protein pilA-N				8.5	GSU1496	Geob3369	Gura2677
20590	geopilin domain 2 protein pilA-C				8.8	GSU1497	−	−
33430	PilZ domain protein				11.0	GSU3028	Geo0640	Gura4082
28450	PilZ domain protein				9.7	GSU1051	Geob0928	Gura0726
17690	PilZ domain protein				10.7	GSU1240	Geob0815	Gura0044
36280	PilZ domain protein				8.6	GSU0137	−	Gura3986
01060	PilZ domain protein				8.2	GSU0078	−	−
37060	PilZ domain protein				10.6	GSU0312	Geob1410	Gura3298
20560	sensor histidine kinase PilS				11.0	−	−	−
20570	sigma-54-dependent transcriptional response pilR				8.3	−	−	−
36190	twitching motility pilus retraction protein pilT-1				8.6	−	−	−
01730	twitching motility pilus retraction protein pilT-3				8.5	−	−	−
20540	twitching motility pilus retraction protein pilT-4				7.4	−	−	−
26020	type IV pilus assembly lipoprotein PilP				5.0	GSU2029	Geob3067	Gura1813
26080	type IV pilus assembly protein PilY1				10.2	GSU1066	Geob3067	−
20530	type IV pilus biogenesis ATPase PilB				6.7	−	−	−
26050	type IV pilus biogenesis ATPase PilM				6.8	−	−	−
26040	type IV pilus biogenesis ATPase PilM				4.2	GSU3069	Geob3069	Gura1811
26030	type IV pilus biogenesis protein PilO				0.0	GSU2030	Geob3068	Gura1812
20550	type IV pilus inner membrane protein PilC				7.9	−	−	−
26130	type IV pilus minor pilin PilE				10.2	GSU3548	−	−
26010	type IV pilus secretin lipoprotein PilQ				5.8	−	−	−
26160	type IV prepilin-like proteins leader peptide pilD				15.1	GSU2043	Geob3081	Gura1794
Identity (%)	20–30	31–40	41–50	51–60	61–70	71–80	81–90	91–100

Tag indicates the locus tag of coding sequences encoding T4P-related proteins in the genome. The number and color scale for the closest protein in the strain indicate the locus tag and amino acid identity (%), respectively. 1: *Geobacter sulfurreducens* PCA1; 2. *G. daltonii* FRC-32; 3. *G. uraniireducens* Rf4.

## Data Availability

The data presented in this study are available upon request from the corresponding author.

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
