# Peer review of "Polyphasic Characterization of Geotalea uranireducens NIT-SL11 Newly Isolated from a Complex of Sewage Sludge and Microbially Reduced Graphene Oxide"

_microorganisms, 2023, doi:10.3390/microorganisms11020349_

Round 1
Reviewer 1 Report
I enjoyed reading this manuscript. It’s a straightforward story clearly written and presented, I don't have any suggestion to improve it... As not a direct expert in this field, “hydrogel” is new to me, and I suggest the authors to cite add more appropriate citation for this concept : Line 298, add:
Yoshida, N.; Miyata, Y.; Doi, K.; Goto, Y.; Nagao, Y.; Tero, R.; Hiraishi, A. Graphene oxide-dependent growth and self-aggregation into a hydrogel complex of exoelectrogenic bacteria. Sci. Rep. 2016, 6, 21867.
Author Response
We thank the affirmation and comment from the reviewer. We cited the reference as the suggestion by the reviewer as shown in L298 in the revised manuscript.
Reviewer 2 Report
Polyphasic characterization of Geotalea uranireducens NIT-SL11 newly isolated from a complex of sewage sludge and microbially reduced graphene oxide
Introduction:
Authors describe bioelectrochemical systems, including microbial fuel cells (MFCs) and microbial electrolysis cells (MECs), and their application in wastewater treatment. Authors then focus on the anode and the use of graphene oxide (GO), the oxidized form of graphene and its reduction (by microbes) into reduced GO (rGO). GO promotes the growth of selective exoelectrogens [12,13] by serving as the extracellular electron acceptor and self-aggregating into a 3D-conductive hydrogel that embeds exoelectrogens and results in significantly more stable energy production than is the case with graphite felt (GF) [14]. A number of bacterial species are reported to reduce GO including Shewanella, Escherichia and other taxa.
The aims of this study are outlined on page 2 line 48 where it states “In this study, an electrogenic bacterium, designated NIT-SL11, was successfully isolated from the rGO complex that converts sewage to electricity and is described in detail.” “The potential extracellular electron transferring (EET) pathway is also proposed”.
2. Materials and Methods This covers the following:
2.1. Isolation and Growth Conditions of NIT-SL11 55
This includes phylogenetic identification based on sequencing of the 16SrRNA gene amplified from the cell lysate and named strain NIT-SL11.
2.2. Morphological, Physiological, and Biochemical Analyses
This included morphology using EM, Gram stain and check on spore formation (light microscopy) Salinity, temperature and pH tolerance, substrate specificity (electron doners) and electron acceptors.
The production of electric current was evaluated via electrochemical cultivation using a graphite plate inoculated with NIT-SL11 (previously described).
2.3. Chemotaxonomic Analysis (Isoprenoid quinones and cellular fatty acid composition)
2.4. Genetic Characterization (Sequencing and phylogenetic analyses).
3. Results and Discussion
3.1. Isolation of NIT-SL11
Cells found to be Gram-negative, non-spore forming, rod shaped, 0.4 x 1.4 microns (fig 1A). Produce orange coloured biofilm (fig 1B). Fig 1 C shows electric current production following pulses of acetate medium.
3.2. Phylogenetic Identification Based on 16S rRNA Sequencing
Figure 2 shows a phylogenetic tree generated using 16S rRNA gene sequences of members of Geobacteraceae family.
3.3. Physiological and Biochemical Characterization
Table 1 summarises the morphological and physiological properties of strain NIT-SL11 and recognized species of the genus Geotalea. What does “ND” mean? Not done? Not determined? Not defined? Not distinguished? Not detected?
3.4. Chemotaxonomic Characterization and 3.5. General Genomic Features
Data in Table 2 shows comparison of chemotaxonomic and genomic properties of strain NIT-SL11 and recognized species of the genus Geotalea. Figure 3 shows the features of the complete genome of strain NIT-SL11. Circular representation of the genome was generated using Tbtools-â…¡ v1. Rings numbered from the outside to inside are: 1, location of tRNA (red), Transfer messenger RNA (blue), and rRNA (green); 2, gene density; 3, c-type cytochromes (black), and type IV pili (red); 4, G+C skew (red, positive; green, negative); 5, protein coding sequences coloured based on KEGG category; 6, G+C content; 7, links showing repetitive sequence≥95% identity (pink, >500 bp; purple, >2 kbp). KEGG, Kyoto Encyclopedia of Genes and Genomes.
3.6. Putative c-Type Cytochromes. Table 3 shows a list of putative c-type cytochrome proteins present in the NIT-SL11 genome. 1: Geobacter sulfurreducens PCA1; 2. G. daltonii FRC-32; 3. G. uraniireducens Rf4. Tag indicates the locus tag of the coding sequence encoding c-type cytochromes in the genome. CM, cytoplasmic membrance; PP, periplasm; EX, extracellular; (-), unknown. The number and colour scale for the closest protein in the strain indicates the locus tag and amino acid identity (%), respectively.
3.7. Type IV Pilus (T4P)-Related Genes
Table 4: List of T4P assembly-related genes in the NIT-SL11 genome Tag indicates the locus tag of coding sequences encoding T4P-related proteins in the genome. The number and colour scale for the closest protein in the strain indicates the locus tag and amino acid identity (%), respectively. 1: Geobacter sulfurreducens PCA1; 2. G. daltonii FRC-32; 3. G. uraniireducens Rf4
Figure 4: Proposed EET pathway of strain NIT-SL11. OM: outer membrane; CM: cytoplasmic membrane.
3.8. Exoelectrogens that form the rGO complex
4. Conclusions
In this study, Geotalea uraniireducens NIT-SL11 was obtained from a hydrogel of microbially reduced GO and anaerobic sludge. The isolated strain was found to utilize various organic acids as electron donors and respired with GO, electrodes, fumarate, and malate. The analysis of the genome of NIT-SL11 suggests two possible extracellular electron transfer pathways. In the first possible pathway, porin-cytochrome (Pcc) EET pathways that begin with the transfer of electrons from the inner membrane quinone pool to membrane-associated cytochromes Cbcl or ImcH, which transfer it further to periplasmic electron carriers PpcA homologs, and PpcA homologs transfer the electrons from the cytoplasm to OmcB-based or Ext-cluster outer membrane conduits, which finally transfer electrons through the outer membrane. In the second possible pathway, the aromatic amino acid-rich conductive pili transfer electrons directly to oxides distant from the cell surfaces.
This is an interesting paper and the methods employed and the data obtained appear sound.
The paper is very well written and I cannot spot any errors of presentation or wording.
The results are well presented and the conclusions robust.
Author Response
We thank the positive comments from the reviewer.